# Solving Partial Label Learning Problem with Multi-Agent Reinforcement Learning

## Abstract

Partial label learning (PLL) deals with classifications when a set of candidate labels instead of the true one is given for each training instance. As a weakly supervised learning problem, the main target of PLL is to discover latent relationships within training samples, and utilize such information to disambiguate noisy labels. Many existing methods choose nearest neighbors of each partially-labeled instance in an unsupervised way such that the obtained instance similarities can be empirically non-optimal and unrelated to the downstream classification task. To address this issue, we propose a novel multi-agent reinforcement learning (MARL) framework which models the connection between each pair of training samples as a reinforcement learning (RL) agent. We use attention-based graph neural network (GNN) to learn the instance similarity, and adaptively refine it using a deterministic policy gradient approach until some pre-defined scoring function is optimized. Different from those two-stage and alternative optimization algorithms whose training procedures are not end-to-end, our RL-based approach directly optimizes the objective function and estimates the instance similarities more precisely. The experimental results show that our method outperforms state-of-the-art competitors with a higher classification accuracy in both synthetic and real examples.

## 1 Introduction

PLL, also known as superset label learning (Liu & Dietterich, 2014; 2012; Gong et al., 2017), has been extensively studied in the past few decades. As a typical weakly-supervised learning problem, PLL assumes that most of training instances are partially labeled and their ground truth labels are unknown. To be more specific, each instance is associated with a small set of candidate labels including the ground truth. PLL has been widely considered in diverse fields, including web mining (Luo & Orabona, 2010), facial age estimation (Zhang et al., 2016), photograph captioning (Duygulu et al., 2002; Barnard et al., 2003; Berg et al., 2004; Gallagher & Chen, 2007) and image annotation (Cour et al., 2011; Zeng et al., 2013). It is usually much easier to get blurry labels than acquiring exact ground truths, and accurately labeling each instance is costly and labor-intensive. For example, the natural photographs collected in the real world may contain multiple human faces and are often tagged ambiguously with several potential names in the captions. The goal is to precisely match the persons in each images with the names, and learn a robust classification model which can be generalized to unseen instances.

PLL has been widely examined in the past few years. How to distinguish fuzzy labels in training sets and recover their true labels plays an important role in developing efficient and robust PLL methods. One main class of PLL methods, such as LSB-CMM(Liu & Dietterich, 2012), M3PL(Yu & Zhang, 2016) and PL-SVM(Nguyen & Caruana, 2008), directly fits the classifier with traditional machine learning models. These methods ignore the relationships between training instances which leads to unfaithful labelling disambiguation results. Some more advanced methods, such as PL-KNN (Hüllermeier & Beringer, 2006), IPAL (Zhang & Yu, 2015) and PL-LEAF (Zhang et al., 2016), utilize the coorelations between training instances provided by some underlying similarity graph learned in an unsupervised manner. These methods have achieved desirable empirical performance, but still suffer from some common issues. For example, the similarity graph generated via some unsupervised approaches may be non-optimal due to its independence of the main classification task. Another weakness of the graph-based methods is that the label prediction of each test example is voted by its neighbors in the training set which makes the prediction results highly depend on the selection of

neighboring instances. Unfortunately, there is usually no guidance of choosing a proper size of the neighboring set in practice. The prediction results can be significantly biased when some unrelated or wrongly-labelled training instances are included as voters. Although some more recent studies such as AGGD (Wang et al., 2019) employ supervised methods to model the instance similarities, the neighbor selection issue has not been fully addressed as they initialize the neighboring set in an unsupervised way and only update the similarity measurements at some specific locations in the subsequent optimization step which has great limitations. On the other hand, these methods need to limit the total number of neighbors and thus some training instances that are useful in improving the predicting accuracy are ignored. Furthermore, the complicated optimization problem proposed by these graph-based methods are difficult to solve in practice.

In this work, we propose a novel end-to-end **P**artial **L**abel learning method using Multi-Agent **R**einforcement **L**earning, called PLRL. Under the MARL setting, each pair of training instances is treated as an individual agent and its action is defined as the similarity measurement between these two instances. All the $n$ training instances build the node set of an underlying similarity graph, and the learned optimal policy precisely quantifies the closeness between any two nodes, which is the edge weight. Unlike traditional two-stage or alternative optimization approaches, our PLRL method directly optimizes a pre-defined score function and adaptively updates the instance similarities in a supervised way which makes the learned similarity graph more related to the main classification task. Specifically, we use GNNs to learn the similarity graph which maximizes a total reward shared by all the $n^2$ agents. With the similarity graph, an estimated probability distribution over the candidate labels for each test example is obtained through some label propagation strategies. Then we incorporate the estimated probabilities with a kernel ridge regression model to carry out the label disambiguation. The MARL framework together with the classification model is end-to-end trained using policy gradients. The main contributions of this paper are summarized as follows.

- We introduce a novel MARL framework to quantify the similarities between training instances and impose no limits on the number of nearest neighbors.
- Different from traditional two-stage or alternative optimization methods, we employ an end-to-end approach to jointly implement predictions and labelling disambiguation, which makes the learned instance similarities more related to the main classification task.
- We use Policy Gradient to efficiently solve the complicated optimization problem which cannot be easily handled by previous studies.
- Experimental results on both synthetic and real datasets show that our method outperforms existing PLL methods with higher classification accuracies in most scenarios, especially in those cases with imbalanced samples.

## 2 PROBLEM STATEMENT

In PLL, a partially labelled training set $\mathcal{D} = \{(\boldsymbol{x}_i, C_i) \mid 1 \leq i \leq n\}$ is given, where $\boldsymbol{x}_i = (x_{i1}, ..., x_{id})^\top \in \mathcal{X}$ is a $d$- dimensional instance and $C_i \subseteq \mathcal{Y}$ is the candidate label set among which only one label is assumed to be valid. $\mathcal{Y} = \{y_1, y_2, ..., y_q\}$ here denotes a label space with $q$ classes. Let $\mathbf{X} = [\boldsymbol{x}_1, ..., \boldsymbol{x}_n]^\top \in \mathbb{R}^{n \times d}$ be the normalized input data matrix. The target of PLL is to induce a multi-class classifier $g(\cdot) : \mathcal{X} \rightarrow \mathcal{Y}$ using such fuzzy label information $\mathcal{D}$ to precisely classify partially-labelled and unseen instances.

In this work, we consider an undirected weighted graph $\mathcal{G} = (\mathcal{V}, \mathbf{W})$ among instances. $\mathcal{V} = \{\boldsymbol{x}_i | 1 \leq i \leq n\}$ is the set of vertices. $\mathbf{W} = [w_{ij}]_{n \times n}$ refers to the non-negative weight adjacency matrix, where $w_{ij} \in [0, 1]$ measures how close the two instances $x_i$ and $x_j$ are to each other with larger value meaning higher correlation. In this work, a novel RL based approach is used to estimate the weighted adjacency matrix $\mathbf{W}$.

For each training example $(\boldsymbol{x}_i, C_i)$, we aim to generate a normalized real-valued vector $\boldsymbol{f}_i \in \mathbb{R}^q$ where each $f_{ij}$ represents the label confidence of the $j$-th label being the true label of $\boldsymbol{x}_i$. The label confidence vector $\boldsymbol{f}_i$ satisfies the following constraints: (i) $\sum_{j=1}^q f_{ij} = 1$ for any $1 \leq i \leq n$, (ii) $f_{ij} \geq 0$ for any $y_{ij} = 1$, and (iii) $f_{ij} = 0$ for any $y_{ij} = 0$. The second and third constraints indicate the potential ground-truth label resides in the candidate label set. When the label confidence of training set $\mathbf{X}$ is generated as $\mathbf{F} = [\boldsymbol{f}_1, ..., \boldsymbol{f}_n]^\top \in \mathbb{R}^{n \times q}$, a classifier $g(\cdot) : \mathcal{X} \rightarrow \mathcal{Y}$ can be induced based on this disambiguation results.

## 3 A MULTI-AGENT RL FRAMEWORK

In this paper, we model the PLL problems via a MARL framework. Each agent learns the similarity $w_{ij}$ between each pair of training instances by fully exploiting the input features and the labelling uncertainty. To be specific, we consider a multi-agent extension of partially observable Markov games, which is defined by a set of states $\mathcal{S}$ describing the possible configurations of all agents, a set of actions $\mathcal{A}_1, ..., \mathcal{A}_N$ and a set of observations $\mathcal{O}_1, ..., \mathcal{O}_N$ for each agent. Each agent $i$ chooses the action following a policy $\pi_i : \mathcal{O}_i \rightarrow \mathcal{A}_i$. We use neural networks to parameterize the policy of each agent, which outputs the similarity between two corresponding instances. The private observation of each agent $i$ is correlated to the overall state $\mathcal{S}$, such that $o_i : \mathcal{S} \rightarrow \mathcal{O}_i$. Since there are $n$ instances, the total number of agents is $N = n^2$. All the $n^2$ agents are fully cooperative to maximize the total reward $r : \mathcal{S} \times \mathcal{A} \rightarrow \mathbb{R}$. In addition, we define the action-value function $Q^{\boldsymbol{\pi}}(\boldsymbol{s}, a_1, ..., a_N) = \mathbb{E}[R(\boldsymbol{s}, \boldsymbol{a})|\boldsymbol{s}_0 = \boldsymbol{s}, \boldsymbol{a}_0 = \boldsymbol{a}]$, where $\boldsymbol{a} = [a_1, ..., a_N]$ and $\boldsymbol{\pi} = (\pi_1, ..., \pi_N)^\top$. In the context of PLL, we highlight the following specifics:

**State**. The entire state space is constructed by the contexts of $n$ training instances, i.e. $\mathcal{S} = \{\boldsymbol{x}_1, ..., \boldsymbol{x}_n\}$. Each agent only observes the contexts of the two related instances, i.e. $\mathcal{O}_{ij} = \{\boldsymbol{x}_i, \boldsymbol{x}_j\}$.

**Action**. For each agent, its action $a_{ij} \in [0, 1]$ measures the similarity between instance $\boldsymbol{x}_i$ and $\boldsymbol{x}_j$, which represents the edge weight in the similarity graph determined by the policy $\pi_{ij}(\boldsymbol{x}_i, \boldsymbol{x}_j)$. The larger value of $a_{ij}$ implies the higher similarity between $\boldsymbol{x}_i$ and $\boldsymbol{x}_j$. Specifically, it is much likely that two instances should be categorized into two different classes of $\mathcal{Y}$ if $a_{ij} = 0$.

For each agent, we employ a neural network to output the action $a_{ij} \in [0, 1]$ with input being the observation $\mathcal{O}_{ij} = \{\boldsymbol{x}_i, \boldsymbol{x}_j\}$, whose parameters are shared by all the $N$ agents. To ensure that the similarity measurement is undirected, i.e. $a_{ij} = a_{ji}$, we borrow the idea of attention mechanism (Vaswani et al., 2017) and the action can be obtained as follows,

$$a_{ij}(\theta) = \pi_{ij}(\boldsymbol{x}_i, \boldsymbol{x}_j) = \text{Sigmoid}\left((W_Q \phi(\boldsymbol{x}_i))^\top W_K \phi(\boldsymbol{x}_j) + (W_K \phi(\boldsymbol{x}_i))^\top W_Q \phi(\boldsymbol{x}_j)\right) \quad (1)$$

where $\phi : \mathbb{R}^p \rightarrow \mathbb{R}^{p'}$ encodes each $x_i$ to a $p'$-dimensional embedding $\phi(x_i)$ (here we use fully connected layer with ReLU activation function), and $W_Q, W_K \in \mathbb{R}^{q \times p'}$ denote the query and key parameter matrices, respectively. $\theta$ includes all the parameters to be learned. It is obvious that the network architecture in Eq.(1) is permutation invariant, and the output is independent of the order of the two inputs $x_i$ and $x_j$. With all the $N$ actions taken by the $N$ agents at each training epoch, we can build a normalized weighted adjacency matrix $\mathbf{W} = [w_{ij}]_{n \times n}$ with $w_{ij} = \frac{a_{ij}}{\sum_k a_{ik}}$. All weight values in $\mathbf{W}$ are updated during the optimization, except that the diagonals are set to zero to avoid self-loops. Different from previous graph-based methods, we allow all the training instances to involve in the prediction of each test example and their importance is optimized via the MARL design.

**Reward**. Since all the $N$ agents are fully cooperative, we define a single reward function to evaluate the learned similarities $\mathbf{W}$ between the $n$ instances, which is defined as follows,

$$-R = \underbrace{\sum_{j=1}^n \left\| \boldsymbol{f}_j - \sum_{i=1}^n w_{ij} \boldsymbol{f}_i \right\|_2}_{\text{(i): LSE of label confidence}} + \mu \underbrace{\sum_{j=1}^n \left\| g(\boldsymbol{x}_j) - \boldsymbol{f}_j \right\|_2}_{\text{(ii): LSE of classifier}} + \eta \underbrace{\sum_{j=1}^n \left\| \boldsymbol{x}_j - \sum_{i=1}^n w_{ij} \boldsymbol{x}_i \right\|_2}_{\text{(iii): LSE of feature}} - \beta \underbrace{\sum_{j=1}^n \log\left(p(\hat{y}_j | \boldsymbol{x}_j)\right)}_{\text{(iv): log-likelihood}}$$

$$(2)$$

The first three terms in (2) are similar to those given in Wang et al. (2019) while the last term is newly raised. Term (iii) measures how each $x_i$ is represented by all the $n$ training instance based on $\mathbf{W}$. Term (i) is exploiting the smoothness assumption that the graph structure in the feature space should be preserved in the label space. $\mathbf{F}$ here is the label confidence of the $n$ training instances which is a function of $\mathbf{W}$ and can be obtained through disambiguation. For term (ii), we expect to minimize the error of the classifier $g(\cdot)$. As shown by Section 4.2, $g(\cdot)$ is determined by $\mathbf{F}$ and thus can also be treated as a function of $\mathbf{W}$. In particular, we add a log-likelihood term (iv) to facilitate the convergence. $\hat{y}_j$ denotes the label prediction of training instance $\boldsymbol{x}_j$ i.e. $p(\hat{y}_j | \boldsymbol{x}_j) = \max_l f_{jl}$ where $f_{jl}$ is the $(j, l)$-th element of $\mathbf{F}$. Since the four terms in (2) all depend on the weighted adjacency matrix $\mathbf{W}$, the negative reward can serve as an objective function to optimize $\mathbf{W}$.

## 4 METHODOLOGY

In this section, we describe the implementation details of the PLRL algorithm. At each training epoch, we apply a disambiguation method to obtain the label confidence $\mathbf{F}$ which is then used to learn the classifier $g(\cdot)$. With the obtained $\mathbf{F}$ and $g(\cdot)$, we can compute the reward function in (2), using policy gradient to train all the $N$ agents and dynamically updating the weighted adjacency matrix $\mathbf{W}$ until the optimal reward $R^*$ is achieved. More details can be found in Sections 4.1, 4.2 and 4.3. With the optimal adjacency matrix $\mathbf{W}^*$, we can finally predict the class label of each unseen instance using the kernel ridge regression (KRR) with optimal model parameters, which is described in Sections 4.4. The flow chart of PLRL is summarized in Figure 1 and the detailed algorithm is provided in the Supplement A.

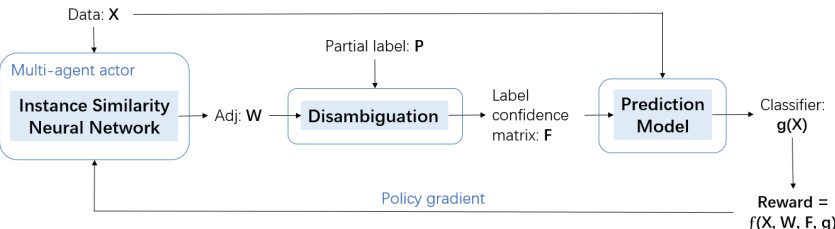

Figure 1: Algorithm architecture of PLRL

### 4.1 COMPUTING $\mathbf{F}$: DISAMBIGUATION

At each training epoch, we disambiguate the candidate labels of training instances using the obtained $\mathbf{W}$ and apply an iterative approach (Zhang & Yu, 2015) to compute the label confidence matrix $\mathbf{F}$ through label propagation. To be specific, we initialize the label confidence matrix $\mathbf{F}^{(0)} = \mathbf{P} = [p_{ij}]_{n \times q}$ based on the partially labelled training set $\mathcal{D}$ before the iteration process starts, such that

$$p_{i,j} = \begin{cases} 1/|C_i|, & \text{if } y_j \in C_i \\ 0, & \text{otherwise} \end{cases} \quad \forall 1 \le i \le n. \tag{3}$$

This initialization step equally distributes the label confidence of $\boldsymbol{x}_i$ over all its candidate classes. Then at the $t$-th iteration, $\mathbf{F}$ is updated by propagating label information along with $\mathbf{W}$, such that

$$\tilde{\mathbf{F}}^{(t)} = \alpha \cdot \mathbf{W}^\top \mathbf{F}^{(t-1)} + (1 - \alpha) \cdot \mathbf{P}, \tag{4}$$

where the hyperparameter $\alpha \in (0, 1)$ balances the contribution of the inherited label information $\mathbf{W}^\top \mathbf{F}^{(t-1)}$ and the initial label confidence $\mathbf{P}$. Then, $\tilde{\mathbf{F}}^{(t)}$ is normalized to get $\mathbf{F}^{(t)}$, i.e.

$$f_{i,j}^{(t)} = \begin{cases} \dfrac{\tilde{f}_{i,j}^{(t)}}{\sum_{y_l \in C_i} \tilde{f}_{i,l}^{(t)}}, & \text{if } y_j \in C_i \\ 0, & \text{otherwise} \end{cases} \quad \forall 1 \le i \le n. \tag{5}$$

After $T$ iterations, we take $\mathbf{F}^{(T)}$ as the final label confidence and adopt the class mass normalization (CMN) (Zhu & Goldberg, 2009) to adjust the disambiguation output towards class prior distribution, which handles unbalanced samples well. We let $\mathbf{F} = \mathbf{M} \circ \mathbf{F}^{(T)}$ where $\circ$ denotes the Hadamard product and $\mathbf{M} = [\boldsymbol{m}, \dots, \boldsymbol{m}]_{n \times q}^\top$ is composed of the vector defined as

$$\boldsymbol{m} = \left[ \frac{m_1}{\hat{m}_1}, \dots, \frac{m_q}{\hat{m}_q} \right]_{q \times 1}^\top, \tag{6}$$

where $m_j = \sum_{i=1}^n p_{i,j}$, $\hat{m}_j = \sum_{i=1}^n f_{i,j}^{(T)}$, $j = 1, ..., q$.

With this disambiguation approach, we get the label confidence matrix $\mathbf{F}$ which is then used to compute terms (i) and (iv) in the objective function (2) to update $\mathbf{W}$.

## 4.2 Obtaining $g(\cdot)$: Kernel Ridge Regression Model

Now, we introduce the classifier which makes predictions for unlabelled instances by using the label confidence $\mathbf{F}$ obtained in Section 4.1. We consider a reproducing kernel Hilbert space (RKHS) and let $\psi(\cdot) : \mathbb{R}^d \to \mathbb{R}^h$ denote the (implicit) nonlinear feature mapping that projects the original feature space to a higher dimensional RKHS space $\mathcal{H}_K$ via a Gaussian kernel function $\kappa(\boldsymbol{x}_i, \boldsymbol{x}_j) = \exp(-\|\boldsymbol{x}_i - \boldsymbol{x}_j\|_2^2/(2\sigma^2))$ where $\sigma$ is the averaged pairwise distances of training instances.

We model the classifier via a kernel regression model using a generalized ridge penalty. As pointed out by Wahba (1990), a general solution to this problem is finite-dimensional, and can be defined as

$$g(\boldsymbol{x}) = \sum_{i=1}^n \boldsymbol{u}_i \kappa(\boldsymbol{x}, \boldsymbol{x}_i) + \boldsymbol{b}, \tag{7}$$

where $\boldsymbol{b}$ and $\boldsymbol{u}_i$, $i = 1, \ldots, n$ are some parameter vectors. Let $\mathbf{K} = \psi(\mathbf{X})\psi(\mathbf{X})^\top$ be a kernel matrix with each element generated by a kernel function $k_{ij} = \kappa(\boldsymbol{x}_i, \boldsymbol{x}_j) = \psi(\boldsymbol{x}_i)^\top \psi(\boldsymbol{x}_j)$ and $\mathbf{U} = [\boldsymbol{u}_1, \ldots, \boldsymbol{u}_n]^\top \in \mathbb{R}^{n \times q}$ contains the combined weights of all the $n$ instances. The classifier can be obtained by fitting the following kernel ridge regression (KRR),

$$\min_{\mathbf{U},\mathbf{b}} \left\| \mathbf{K}\mathbf{U} + \mathbf{1}\mathbf{b}^\top - \mathbf{F} \right\|_{\mathrm{F}}^2 + \lambda \operatorname{tr}\left( \mathbf{U}^\top \mathbf{K} \mathbf{U} \right), \tag{8}$$

where $\mathbf{1}$ denotes a vector of 1's, $\| \cdot \|_{\mathrm{F}}$ represents the Frobenius norm, and $\operatorname{tr}(\cdot)$ is the trace. Let the gradients of (8) with respect to $\mathbf{U}$ and $\mathbf{b}$ be zeros, we can obtain the solution of $\mathbf{U}$ and $\mathbf{b}$ as follows,

$$
\begin{aligned}
\mathbf{U} &= \left( \mathbf{K} + \lambda \mathbf{I} - \frac{\mathbf{1}\mathbf{1}^\top \mathbf{K}}{n} \right)^{-1} \left( \mathbf{F} - \frac{\mathbf{1}\mathbf{1}^\top \mathbf{F}}{n} \right), \\
\mathbf{b} &= \frac{1}{n} \left( \mathbf{F}^\top \mathbf{1} - \mathbf{U}^\top \mathbf{K}^\top \mathbf{1} \right),
\end{aligned}
\tag{9}
$$

which are then used to compute term (ii) in the objective function (2).

## 4.3 Updating $\mathbf{W}$: Policy Gradient

At each training epoch, the agent $i, j$ observes its own observation $\mathcal{O}_{ij}$ and outputs the similarity $w_{ij}$ between $\boldsymbol{x}_i$ and $\boldsymbol{x}_j$. With the $N$ obtained $w_{ij}$'s, we carry out the disambiguation step in Section 4.1 and calculate the prediction model given in Section 4.2. Then the reward $R$ can be obtained.

Let the expected reward $J(\boldsymbol{\pi}_\theta) = \mathbb{E}_{s \in \mathcal{S}, a \sim \boldsymbol{\pi}_\theta}[R]$ be the objective function of PLRL, where the detailed formulation of $R$ is given by (2) and the computations of $\mathbf{F}$ and $g(\cdot)$ are described in Sections 4.1 and 4.2. All the four components in (2) work together and jointly affect the estimation of $\mathbf{W}$, which makes the learned similarity graph related to the classification task, and thus the neighboring information can be well utilized to improve the prediction accuracy. The gradient $\nabla_\theta J(\boldsymbol{\pi}_\theta)$ is obtained by the deterministic policy gradient (DPG) algorithm (Silver et al., 2014) as

$$\nabla_\theta J(\boldsymbol{\pi}_\theta) = \nabla_\theta \boldsymbol{\pi}_\theta(\boldsymbol{s}) \nabla_{\boldsymbol{a}} Q_{\boldsymbol{\pi}}(\boldsymbol{s}, a_1, \ldots, a_N)|_{a_j = \pi_j(o_j), j=1,\ldots,N}, \tag{10}$$

where $\boldsymbol{s} = \mathbf{X}$, $o_{il} = \{\boldsymbol{x}_i, \boldsymbol{x}_l\}$, $1 \le i, l \le n$. In practice, we can pre-train the model and properly initialize the adjacency matrix $\mathbf{W}$, which helps shorten the training process of the PLRL algorithm. A good starting point of the matrix $\mathbf{W}$ can be obtained by using some existing unsupervised methods such as $k$-NN (Zhang & Yu, 2015). Then the proposed PLRL algorithm can further improve the prediction accuracy by refining the matrix $\mathbf{W}$ in a supervised way.

## 4.4 Prediction

As the DPG algorithm converges, we can get the optimal similarity graph $\mathbf{W}^*$ and the optimal label confidence matrix $\mathbf{F}^*$. The finally prediction of each unseen instance $\boldsymbol{x}^*$ is given by

$$y^* = \arg\max_j \sum_{i=1}^n u_{ij}^* \kappa(\boldsymbol{x}^*, \boldsymbol{x}_i) + b_j^*. \tag{11}$$

where $u_{ij}^*$ is in the $i$-th row, $j$-th column of $\mathbf{U}^*$ and $b_j^*$ is the $j$-th element in $\mathbf{b}^*$. Both $\mathbf{U}^*$ and $\mathbf{b}^*$ here can be obtained by (9) using the optimal $\mathbf{W}^*$ and $\mathbf{F}^*$.

## 5 EXPERIMENTS

In this section, we conduct both synthetic and real-world experiments to demonstrate the advantages of the proposed PLRL algorithm in solving PLL problems. PLRL is compared with seven SOTA PLL algorithms, whose parameters are fine-tuned as suggested by the literature. The configurations of these methods are summarized in Table 1 and the details are provided in the Supplement B. For PLRL, the optimal choices of the regularization coefficients are $\mu = 1, \eta = 0.5$ and $\beta = 0.05$ according to the cross-validation results. More analysis about the parameter sensitivity are in the Supplement E.1. For each method, we perform five-fold cross-validation, and report the mean accuracy together with their standard deviations. In addition, we apply the $t$-test at 0.05 significance level to assess the performance of the proposed method.

Table 1: Comparing methods

| Methods | Graph-based | Configuration |
|---------|-------------|---------------|
| PLRL (Ours) | Yes | $\alpha = 0.95, T = 100, \lambda = 0.05, \mu = 1, \eta = 0.5, \beta = 0.05$ |
| AGGD (Wang et al., 2019) | Yes | $k = 10, T = 10, \lambda = 1, \mu = 1, \gamma = 0.05$ |
| IPAL (Zhang & Yu, 2015) | Yes | $k = 10, \alpha = 0.95, T = 100$ |
| PL-KNN (Hüllermeier & Beringer, 2006) | Yes | $k = 10$ |
| SURE (Feng & An, 2019a) | No | $\lambda = 0.5, \beta = 0.05$ |
| LSB-CMM (Liu & Dietterich, 2012) | No | $\sigma^2 = 1, K = 80, \alpha = 0.05$ |
| CLPL (Cour et al., 2011) | No | SVM with the squared hinge loss |
| PL-SVM (Nguyen & Caruana, 2008) | No | $\lambda = 0.01$ |

Our implementation is based on PyTorch (Paszke et al., 2019) and all the experiments were carried out with NVIDIA Tesla P100 GPUs. When handling large-scale datasets, we employ a batch training strategy to reduce the influence of the limited memory. To be specific, we randomly split the whole dataset into several parts and iteratively feed them into the GPUs when training the GNN model. Then we use the trained model to make predictions for all training instances. In practice, our PLRL method takes nearly the same computation time as other recently proposed parametric methods including SURE and AGGD. For the small datasets with sample size less than 2000, the whole training procedure takes about 1 to 10 minuets, and for some large datasets with more than 10000 samples, it takes about 1 to 3 hours. Considering the performance gain of PLRL in practice, the training cost is acceptable.

### 5.1 SIMULATION STUDY

To show that the MARL design can better capture the instance similarities and thus improve the prediction accuracy, we design a simulation experiment and compare our PLRL algorithm with three SOTA graph-based methods, AGGD, IPAL and PL-KNN.

We let the sample size $n$ be 110, the dimension of features $d$ be 6, the number of classes $q$ be 5, and the number of wrong labels in each candidate set be $r \in \{1, 2, 3\}$. Each of the six instance features is generated from a Gaussian distribution, an uniform distribution or a binomial distribution. We randomly divided the $n$ samples into $q$ classes, and the features of instances belonging to different classes follow different distributions. The detailed distribution settings are summarized in Table 7 of the Supplements. The data generating procedure is given as follows,

1. Randomly assign one of the $q$ class labels to each of the $n$ instances as the true label $y$.

2. For each instance, generate data features $\mathbf{X}^{(y)}$ according to Table 7 of the Supplements.

3. We consider an underlying connective graph and simulate the adjacency matrix $\bar{\mathbf{A}} = [\bar{a}_{ij}]_{n \times n}$ by $\bar{a}_{ij} = \mathbf{1}_{\{y_i = y_j\}} \mathbf{1}_{\{\kappa_{ij} > 0.75\}}$ for $1 \leq i, j \leq n$, which measures the similarities between the $n$ training instances. $\kappa_{ij}$ here is the Gaussian kernel $\kappa(\boldsymbol{x}_i, \boldsymbol{x}_j) = \exp(-\|\boldsymbol{x}_i - \boldsymbol{x}_j\|_2^2/(2\sigma^2))$ where $\sigma$ is the averaged pairwise distances of all instances.

4. For each instance, simulate the $r$ partial labels by randomly selecting $r$ labels from the $q - 1$ candidate labels that are not equal to $y$.

We repeat this data generating procedure 40 times and conduct five-fold cross-validation each time to report the mean prediction accuracy. Due to the small sample size of the generated dataset, we set the total number of training epochs to be 200 for each method.

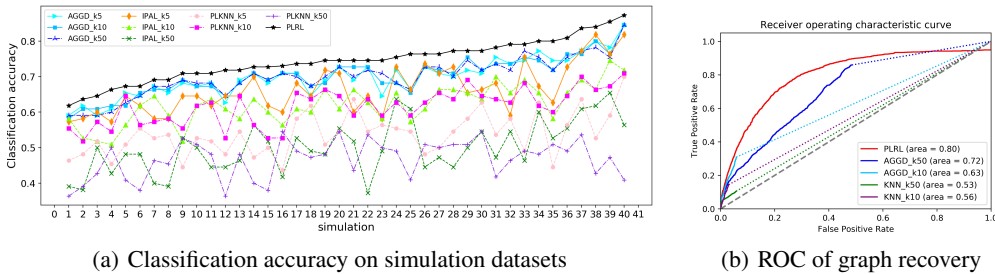

(a) Classification accuracy on simulation datasets (b) ROC of graph recovery

Figure 2: Classification and graph recovery performance on simulation datasets

For the three competitors, we plot their prediction results under different choices of $k$, which is the total number of neighbours of each training instance when building the underlying similarity graph. The prediction performance of all the four methods is visualized in Figure 2(a), where the 40 replicates are sorted in an ascending order according to the classification accuracy. As Figure 2(a) shows, PLRL performs consistently well while the classification accuracy of the the other three varies a lot with the $k$ selections across the 40 replicates. In some cases a small neighbor set is preferred while in other cases a denser similarity graph may be helpful. Since the optimal number of $k$ is usually unknown in the real world, the performance of the three graph-based methods can be highly affected with a improperly selected $k$. Our method addresses this issue by allowing all the $n^2$ edges to have non-zero weights which are optimized by the end-to-end MARL design.

We also show how each method recovers the true underlying graph $\bar{\mathbf{A}}$. To be specific, we transform each obtained weighted adjacency matrix $\mathbf{W}$ into an unweighted one $\hat{\mathbf{A}} = [\hat{a}_{ij}]_{n \times n}$ where each $\hat{a}_{ij} = \mathbf{1}_{\{w_{ij} > \tau\}}$ and $\tau \in (0, 1)$ here represents a truncation threshold. In Figure 2(b), we draw five ROC curves based on the learned weighted graph obtained by the five methods. It should be noted that both IPAL and PL-KNN estimate the smilarity graph using K Nearest Neighbor and thus we use "KNN" to represent them. As Figure 2(b) shows, PLRL can better estimates the true graph. For both AGGD and KNN, the estimation performance does not improve when a certain TPR is reached.

Table 2 summarizes the graph estimations under $\tau = 0.05$. F-norm denotes the Frobenius distance between the true graph $\bar{\mathbf{A}}$ and the estimated graph $\mathbf{W}$. Higher values of NNZ, TPR, and smaller values of FDR, SHD and F norm distance demonstrate that PLRL can discover more true edges than the other three and achieve the best performance in recovering the underlying graph. The simulation study indicates that PLRL can better capture the similarities between training instances which helps improve the final prediction accuracy.

Table 2: Graph recovery performance on one simulation dataset

|  | FDR | TPR | SHD | NNZ | F norm |
|---|---|---|---|---|---|
| PLRL | 0.314 | 0.212 | 1052 | 384 | 4.74 |
| AGGD($k$=5) | 0.347 | 0.157 | 1101 | 296 | 5.66 |
| AGGD($k$=10) | 0.384 | 0.174 | 1110 | 323 | 5.30 |
| AGGD($k$=50) | 0.411 | 0.174 | 1126 | 349 | 5.28 |
| KNN($k$=5) | 0.388 | 0.154 | 1121 | 297 | 5.48 |
| KNN($k$=10) | 0.489 | 0.164 | 1180 | 375 | 4.84 |
| KNN($k$=50) | 0.681 | 0.092 | 1311 | 334 | 5.49 |

## 5.2 CONTROLLED UCI DATASETS

Following the common design of previous PLL studies, we generate an artificial partially labelled dataset based on the UCI dataset (Dua & Graff, 2019). The characteristics of eight UCI datasets are summarized in Table 8 of the Supplements. We set the proportion of PL examples $p = 1$ across all

our experiments, and vary the number of wrong labels $r$ and the probability one specific false positive label co-occurs with the true label $\epsilon$. The configuration of the two settings is provided as follows: (I) $p = 1, r = 1, \epsilon \in \{0.2, 0.3, ..., 0.8\}$. (II) $p = 1, r \in \{1, 2, 3, 4, 5, 6, 7\}$.

The details of the experiment performance are shown in Supplement D. To clearly illustrate the advantage of the proposed PLRL algorithm, we report the win/tie/loss results between PLRL and each competing method using two-sample $t$-test. As Table 3 shows, PLRL significantly outperforms its competitors with win/tie rates greater than $93.0\%$ among 86 set-ups. In particular, for some recently proposed methods such as SURE and AGGD, PLRL is superior or comparable to them in most cases.

Table 3: Win/tie/loss (pairwise $t$-test at 0.05 significance level) counts on the controlled UCI datasets between PLRL and the comparing algorithms

|                    | SURE     | AGGD     | IPAL    | CLPL    | PL-SVM  | PL-KNN  | LSB-CMM |
|--------------------|----------|----------|---------|---------|---------|---------|---------|
| config1($\epsilon$) | 24/27/1  | 25/26/1  | 39/9/4  | 52/0/0  | 52/0/0  | 47/5/0  | 35/12/5 |
| config2($r$)        | 13/21/0  | 10/24/0  | 27/5/2  | 34/0/0  | 34/0/0  | 30/4/0  | 23/11/0 |
| In total           | 37/48/1  | 35/50/1  | 66/14/6 | 86/0/0  | 86/0/0  | 77/9/0  | 58/23/5 |
| PLRL win/tie rate  | 98.8%    | 98.8%    | 93.0%   | 100.0%  | 100.0%  | 100.0%  | 94.2%   |

## 5.3 REAL-WORLD DATASETS

We use five real-world datasets to validate the proposed method, including Lost (Cour et al., 2011), Soccer Player (Zeng et al., 2013), Yahoo! News (Guillaumin et al., 2010), MSRCv2 (Liu & Dietterich, 2012) and BirdSong (Briggs et al., 2012). The first three are for automatic face naming, the fourth is for object classification, and the last is to classify bird songs. The characteristics of these five datasets are summarized in Table 9 of the Supplements, as well as the average number of candidate labels.

The mean inductive classification accuracy with its standard deviation for each algorithm on unlabelled test data are summarized in Table 4. Pairwise $t$-tests at 0.05 significance level is conducted based on 5-fold cross-validation. As shown in Table 4, PLRL significantly outperforms the others in four datasets and achieves the second best prediction accuracy in Yahoo! News, which indicates that PLRL is less sensitive to the data structures and performs consistently well in practice.

Table 5 presents the transductive classification accuracy on partially labelled training samples, which reflects the disambiguation capacity of each method in recovering ground-truths from the candidate label set. For PLRL, SURE and AGGD, the generated label confidence vector $\boldsymbol{f}_i$ can be used to predict the ground-truth label for each partially labelled training instance $\boldsymbol{x}_i$ such that $\hat{y}_i = \arg\max_{y_k \in C_i} \boldsymbol{f}_{ik}$. Other approaches directly make predictions by choosing the most likely $\hat{y}_i \in C_i$.

As shown in Table 5, the performance of PLRL is superior to other algorithms in terms of a higher transductive prediction accuracy. In particular, PLRL significantly outperforms CLPL, IPAL, PL-KNN, and LSB-CMM in all the five real-world datasets. When compared to SURE, AGGD and PL-SVM, PLRL can still achieves the best performance in most cases. These results imply that PLRL performs better in disambiguating blurry labels and can extract more useful information which helps to build a more precise prediction model.

Table 4: Inductive classification accuracy (mean&std) of each method on the real-world partial label datasets, where ●/○ indicates whether the performance of PLRL is statistically superior/inferior to the comparing algorithm on each data set (pairwise $t$-test at 0.05 significance level)

|         | Lost           | MSRCv2         | BirdSong       | Soccer Player  | Yahoo! News    |
|---------|----------------|----------------|----------------|----------------|----------------|
| PLRL    | **0.810 (0.038)** | **0.585 (0.034)** | **0.749 (0.013)** | **0.557 (0.007)** | 0.652 (0.012)  |
| SURE    | 0.765 (0.022)● | 0.463 (0.034)● | 0.733 (0.020)  | 0.529 (0.011)● | 0.628 (0.011)● |
| AGGD    | 0.758 (0.029)● | 0.494 (0.024)● | 0.731 (0.013)  | 0.538 (0.010)● | 0.645 (0.009)  |
| IPAL    | 0.688 (0.033)● | 0.522 (0.027)● | 0.708 (0.011)● | 0.547 (0.006)  | **0.659 (0.008)** |
| CLPL    | 0.728 (0.041)● | 0.412 (0.024)● | 0.631 (0.010)● | 0.366 (0.006)● | 0.461 (0.008)● |
| PL-SVM  | 0.693 (0.051)● | 0.467 (0.055)● | 0.629 (0.051)● | 0.451 (0.007)● | 0.544 (0.164)● |
| PL-KNN  | 0.342 (0.024)● | 0.437 (0.018)● | 0.646 (0.013)● | 0.493 (0.010)● | 0.402 (0.003)● |
| LSB-CMM | 0.652 (0.033)● | 0.462 (0.035)● | 0.684 (0.023)● | 0.491 (0.011)● | 0.522 (0.007)● |

Table 5: Transductive classification accuracy (mean&std) of each method on the real-world partial label datasets

|  | Lost | MSRCv2 | BirdSong | Soccer Player | Yahoo! News |
|---|---|---|---|---|---|
| PLRL | **0.907 (0.006)** | **0.781 (0.005)** | **0.867 (0.003)** | 0.708 (0.004) | **0.866 (0.002)** |
| SURE | 0.828 (0.012)● | 0.608 (0.009)● | 0.838 (0.016)● | 0.703 (0.001) | 0.812 (0.001)● |
| AGGD | 0.866 (0.004)● | 0.637 (0.011)● | 0.840 (0.004)● | 0.717 (0.003) | 0.850 (0.002)● |
| IPAL | 0.834 (0.007)● | 0.698 (0.013)● | 0.832 (0.004)● | 0.679 (0.002)● | 0.850 (0.001)● |
| CLPL | 0.877 (0.002)● | 0.595 (0.010)● | 0.762 (0.006)● | 0.669 (0.005)● | 0.768 (0.004)● |
| PL-SVM | 0.861 (0.020)● | 0.652 (0.040)● | 0.778 (0.044)● | **0.722 (0.004)**○ | 0.790 (0.108)● |
| PL-KNN | 0.614 (0.015)● | 0.598 (0.008)● | 0.757 (0.008)● | 0.568 (0.001)● | 0.689 (0.002)● |
| LSB-CMM | 0.797 (0.008)● | 0.613 (0.016)● | 0.814 (0.013)● | 0.677 (0.003)● | 0.821 (0.012)● |

## 5.4 FURTHER STUDY

To fairly evaluate the contributions of the MARL design, the GNN model used to learn the weighted graph and the KRR model, we carry out an ablation study on four different datasets to quantitatively illustrate their necessity. Specifically, we compare the full PLRL model with the ones removing either GNN&RL or KRR and also the baseline model IPAL, whose detailed architectures are described as follows,

- "w/o GNN&RL" represents the prediction method that uses $k$-NN to estimate the underlying graph and make predictions via KRR. In this case, both GNN and RL are removed.
- "w/o KRR " refers to the case that the reward is obtained by $\sum_{j=1}^{n} \left\| \boldsymbol{f}_j - \sum_i w_{ij} \boldsymbol{f}_i \right\|_2 + \eta \sum_{j=1}^{n} \left\| \boldsymbol{x}_j - \sum_i w_{ij} \boldsymbol{x}_i \right\|_2$. Since $g(\cdot)$ is removed, the final prediction of each test instance is voted by it neighbors.
- IPAL serves as the baseline of PLRL without KRR and GNN&RL.

Table 6: Prediction accuracy of PLRL and its ablated variants on four partial label datasets

|  | vehicle ($r$=2) | steel ($r$=2) | Lost | MSRCv2 |
|---|---|---|---|---|
| PLRL | **0.732 (0.042)** | **0.737 (0.024)** | **0.810 (0.038)** | **0.585 (0.034)** |
| w/o GNN&RL | 0.710 (0.043)● | 0.723 (0.018) | 0.764 (0.016)● | 0.543 (0.029)● |
| w/o KRR | 0.710 (0.066)● | 0.684 (0.016)● | 0.741 (0.022)● | 0.573 (0.022) |
| IPAL | 0.677 (0.067)● | 0.686 (0.014)● | 0.688 (0.033)● | 0.522 (0.027)● |

As shown in Table 6, the full PLRL model consistently outperforms the other three in most scenarios with statistically significant $t$-test results. The prediction accuracy decreases when either the KRR model or the GNN&RL mechanism is removed. Thus, both of these two components are important in improving the baseline performance although their contributions can be varied in different datasets, which validates their effectiveness and indispensability.

We also show that the reward and the classification accuracy on both training and test datasets converge after 1000 epochs. More details can be found in the Supplement E.2.

## 6 CONCLUSION

In this work, we propose a novel PLL method, called PLRL, which models the instance similarities using an MARL based GNN model and enhances the disambiguation capacity. Our method takes all training samples into consideration when building the similarity graph and can better utilize the neighboring information. Thanks to the end-to-end design, the graph structure learned by PLRL is more related to the main classification task.

Despite the empirical success PLRL achieves, there are still some open questions to be answered. First, we need to figure out why the improvement of PLRL over baselines is varied across different datasets. Second, the significance of the improvement by PLRL largely lies in the usage of GNN and RL. It may bring some new ideas to other weakly supervised learning problems such as graph estimations and link predictions.

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
