# OpenReview forum: "Solving Partial Label Learning Problem with Multi-Agent Reinforcement Learning"
_ICLR.cc/2023/Conference — Submitted to ICLR 2023_

### Official Review · Reviewer_cHoo · 2022-10-24

**Confidence:** 4
**Correctness:** 4
**Technical Novelty And Significance:** 2
**Empirical Novelty And Significance:** 2
**Recommendation:** 5

**Clarity, Quality, Novelty And Reproducibility:**

Novelty: Good. paper makes non-trivial advances over the current state-of-the-art.


Quality: Good. The paper appears to be technically sound. The proofs, if applicable, appear to be correct, but I have not carefully checked the details. The experimental evaluation, if applicable, is adequate, and the results convincingly support the main claims.

Clarity: Good. The paper is well organized but the presentation has minor details that could be improved.

Reproducibility: Good. Key resources (e.g., proofs, code, data) are available and sufficient details (e.g., proofs, experimental setup) are described such that an expert should be able to reproduce the main results.


**Strength And Weaknesses:**

Strengths:
1. Based on instance-based disambiguation, the authors propose a novel learning framework by introducing a multi-agent reinforcement learning approach. Although an additional computational effort is introduced, it is worthwhile from the experimental results.
2. Technical steps and experimental setup are clearly explained. Easy to reproduce experimental results.
3. The paper is well-organized and easy to understand.

Weaknesses:
1. The method proposed in this paper lacks effective theoretical support. Effectiveness is an attempt based on heuristics and experience.
2. There are a few comparison methods in the experimental part. To my knowledge, many state-of-the-art methods have not appeared in the paper. To demonstrate the effectiveness of the proposed method, experiments should be conducted in this paper to compare with recent methods [1].
3. Compared with the traditional instance-based PLL, the main contribution of this paper is the improvement in the estimation of the similarity between instances. The article lacks an explanation for the plausibility of this improvement or how it is better than other estimation methods.

[1] PiCO: Contrastive Label Disambiguation for Partial Label Learning. In ICLR, 2022.


**Summary Of The Paper:**

Partial label learning (PLL) is a typical weakly supervised learning, and there are many solutions. This paper proposes a novel instance-based multi-agent reinforcement learning framework called PLRL, which uses an attention-based graph neural network (GNN) to learn instance similarity. Different from the commonly used two-stage or alternative optimization methods, the authors use the RL-based approach to directly optimize the objective function to improve the accuracy of similarity estimation between instances. Experimental results on the datasets that are frequently used in PLL demonstrate this work is effective.

**Summary Of The Review:**

This paper mainly proposes a novel multi-agent reinforcement learning framework to solve PLL. In the instance-based method, The author introduced a more complex and effective similarity calculation model and obtained better results. Experiments show that the proposed method is effective. It is a continuation of research in one or more areas of AI. Marginally above the acceptance threshold.

---

### Official Review · Reviewer_UdWa · 2022-11-03

**Confidence:** 2
**Correctness:** 2
**Technical Novelty And Significance:** 3
**Empirical Novelty And Significance:** 3
**Recommendation:** 3

**Clarity, Quality, Novelty And Reproducibility:**

Method is unclear to me.

Novelty - method lacks a related work section so can not easily evaluate with respect to the larger field.



**Strength And Weaknesses:**

## Strengths
1) Strong experimental results against established datasets
2) Really like the statical significance used to evaluate over other methods.

## Weaknesses

1) The authors assert we have $n^2$ agents (where $n$ refers to the number of instances). Doesn't this mean the number of agents groups QUADRATICALLY with the dataset size?
2) Its unclear if this is MARL? There is some redundancy in your method, if each agent only sees one instance, why  not have a single agent see all instances. The redundancy is clear from the notation in eq1 :$πij(xi,xj)$
3) Its unclear how this is training end-to-end, the reinforce update for the agent is not done in the same gradient update as that for the GNN

**Summary Of The Paper:**

The paper proposes an end-to-end solution for partial label learning, in which classification must be done on data which provides a set of labels during training. The main difficulty in existing approaches are that they either treat this as purely classification task (losing any similarity) or apply a  two step approach - of developing a similarity graph first then training over this (non-direct optimisation).

The authors propose Partial Label learning method using Multi-Agent Reinforcement Learning (PLRL).

This treats every edge (2 data points) as an agent, with the single (continuous) action of producing the similarity for that edge. These agents are trained in a fully cooperative setting. The reward function is conditioned on a GNN + Kernel Method which is trained in tandem. Each agent is a separate policy and only acts for one action step.

My generous interpretation is that the GNN+Kernal approach is being distilled in n-agents which are hopefully better at generalising at test time.

**Summary Of The Review:**

Paper has strong experimental results but a very unclear method section. The use of multi-agent is not well understood or conveyed. The quality is below the acceptance threshold

---

### Official Review · Reviewer_h8m7 · 2022-11-03

**Confidence:** 3
**Correctness:** 1
**Technical Novelty And Significance:** 3
**Empirical Novelty And Significance:** 3
**Recommendation:** 3

**Clarity, Quality, Novelty And Reproducibility:**

Clarity: Low. It is unclear why the authors are claiming to use multi-agent RL when this is not necessary.

Quality: Low. There are some fundamental flaws with the manuscript, as stated above.

Novelty: High.

Reproducibility: Medium. The code is not included as part of the submission.

**Strength And Weaknesses:**

## Strengths

- The idea of using RL for solving partial-label learning is interesting and novel.

## Weaknesses

I think there are some fundamental flaws in this manuscript.

- Firstly, the setting that the authors introduce does not seem to have the notion of time steps. The agent(s) receive a context and then after performing an action, get an immediate return (using the hand-designed reward function). This seems like a contextual multi-armed bandit problem (one-step MDP), rather than an RL problem.
- Next, I don’t quite understand why the agents try to pose this as a *multi-agent* RL problem. There are no inherent agencies here and neither there is a method here than can be applied to other MARL settings. It seems like the authors artificially decided to call a pair of vertices “agents” and their method, therefore, becomes a multi-agent RL algorithm. There are no restrictions in this problem to make it multi-agent (e.g. partial observability or communication constraints).

I believe this paper requires a major rewriting and to be translated into “Solving Partial Labeling Learning Problem with Multi-Armed Bandit”.

**Summary Of The Paper:**

Discovering latent relationships within training samples is an important problem that has been studied in depth before. In this work, the authors propose solving the partial label learning (PPL) problem with multi-agent reinforcement learning (RL). Their solution uses an attention-based graph neural network (GNN) to learn the instance similarity, and adaptively refine it using a deterministic policy gradient approach until some pre-defined scoring function is optimized. The empirical results suggest that the new method outperforms existing baselines with higher classification accuracy in both synthetic and real examples.

**Summary Of The Review:**

The authors use an interesting idea of applying RL for solving the partial label learning problem. However, I believe there are fundamental flaws with the manuscript, as described above.

---

### Decision · Program_Chairs · 2023-01-20

**Decision:**

Reject

**Justification For Why Not Higher Score:**

Unanimous agreement among reviewers and no author rebuttal.

**Justification For Why Not Lower Score:**

N/A

**Metareview: Summary, Strengths And Weaknesses:**

The reviewers unanimously agree that this paper is not ready for publication. Therefore I recommend rejection.